# Subsidy Accountability and Biodiversity Loss Drivers: Following the Money in the Chilean Silvoagricultural Sector

## Cristian Pérez [1,2,*] and Javier A. Simonetti [1]

1. Departamento de Ciencias Ecológicas, Facultad de Ciencias, Universidad de Chile, Las Palmeras 3425, Ñuñoa 7800003, Santiago, Chile
2. Programa de Doctorado en Ciencias Silvoagropecuarias y Veterinarias, Campus Sur Universidad de Chile, Santa Rosa 11315, La Pintana 8820808, Santiago, Chile
* Correspondence: cperezm@veterinaria.uchile.cl

**Abstract:** In Chile, promotion of activities in the silvoagricultural sector has been made through the implementation of Instruments of Productive Promotion, which are governmental interventions oriented to increase productive systems by applying economic incentives. However, its use has not been exempted of criticism due to the poor articulation and coordination between the programs and because their implementation has lacked a coordinated territorial approach. Chile has committed to different international frameworks to protect biodiversity, including the Convention on Biological Diversity that, through the Aichi targets, aimed to either eliminate or reform incentives, including subsidies, to minimize negative impacts and to manage agriculture in a sustainable manner. Allocation of IPPs used to finance field work (IPP-FFWs) at the silvoagricultural sector was analyzed, including amounts granted, use of the funds, and geographical distribution; they explored eventual links with biodiversity trends, including identified drivers of biodiversity loss. We found that, in the last two decades, IPP-FFWs have more than quintupled; the activities funded relate to main anthropogenic factors associated with ecosystems deterioration, including land use change and plantations with exotic species; the funding mostly occurs where most relevant Chilean terrestrial biodiversity features concentrate and where most ecosystems that have been classified under risk are located.

**Keywords:** biodiversity; Chile; incentives; instruments of productive promotion; silvoagricultural systems; subsidies

## 1. Introduction

For nearly a century, governments around the world have subsidized their agricultural sectors to a degree that has few parallels [1]. Such subsidies are economic interventions and policies oriented to protect the agricultural sector, assuming that they might release farming from constraints due to rural market imperfections, enhancing their agricultural productivity [2]. Subsidies include money transfers, payments, support, assistance, and aid [3].

While it has been reported that subsidies might foster the agricultural sector, acting as catalysts towards the adoption of new technologies [4] and improving technical efficiency [5], increasing productivity as a consequence [6], detrimental effects of their use have also been reported since the late 1980s, ranging from market distortions to environmental impacts [7–9]. On environmental terms, agricultural subsidies intensify agricultural production, require higher use of pesticides and fertilizers, increase land use change and degradation, increase water pollution, and lead to biodiversity loss [10]. In fact, agricultural intensification causes negative impacts on the environment and biodiversity, including increased erosion, lower soil fertility, and an extensive conversion of land use with the loss of natural habitats [11,12], and it is among the leading causes of water pollution [13]. Moreover, the biological impoverishment associated with agricultural intensification may compromise the delivery of ecosystem services important for human welfare [14]. Even

considering this, subsidies are still common practice in most countries worldwide, accounting for over USD 700 billion per year, despite often driving environmental damage and failing to provide social benefits beyond farming [15].

At the global level, the Convention on Biological Diversity (CBD), through its Strategic Plan for Biological Diversity 2011–2020 (Aichi targets), established that incentives, including subsidies, harmful to biodiversity ought to be eliminated, phased out, or reformed to minimize or avoid negative impacts [16]. Similarly, the Sustainable Development Goals of the United Nations Development Program aim to gradually eliminate detrimental subsidies [17]. Additionally, the Aichi targets also established that areas under agriculture, aquaculture, and forestry ought to be managed sustainably, ensuring the conservation of biodiversity. To fulfill such an objective, harmful subsidies ought to be eliminated or reformed. Moreover, given the fact that the deadline for the 2020 Aichi targets is over, and considering the relevance of the subject, parties and observers to the CBD suggested that the post-2020 global biodiversity framework should include a reference to the fact that subsidies in productive sectors (agriculture, fisheries, forestry, etc.) harmful to biodiversity are eliminated by 2030 [18]. Harmful subsidies are defined as "a result of a government action that confers an advantage on consumers or producers, in order to supplement their income or lower their costs, but in doing so, discriminates against sound environmental practices" [19] (p. 16).

In Chile, among other purposes, Instruments of Productive Promotion (IPPs hereafter) are used to increase, improve, and diversify agricultural activities [20,21]. In addition, to increase productivity associated to their use, IPPs have also caused expansion of the land used, as well as increased the inputs of agrochemicals, machinery, and fuel, which are all factors that have resulted in environmental impacts at the local, regional, and global level [22].

The reliance on IPPs for agricultural promotion in Chile has not been exempted of criticism due to the poor articulation and coordination of the subsidies offered, including the lack of a territorial strategic approach [21]. In fact, neither criteria nor direct indicators to evaluate their outputs have been developed to determine both the quantitative and qualitative impacts of their implementation [23]. In this context, the Office of Agricultural Studies and Policies of the Ministry of Agriculture [24], as part of the challenges of the Chilean agriculture towards 2030, indicates that the coordination and the linkage of the instruments of promotion used by the state are considered as pivotal to address the challenges of an environmentally sustainable silvoagricultural production.

Regarding the achievement of Aichi target 3, associated with the elimination, phasing out, or minimization of detrimental subsidies, evaluations by the Chilean Ministry of the Environment in 2014 and 2019 evidenced no registry of detrimental subsidies, attesting that progress towards its fulfillment is either progressing insufficiently or without significant change [25,26].

In economic terms, since 2013 to 2021, the contribution to the Gross Domestic Product (GDP) of the silvoagricultural sector, understood as forestry and agricultural activities (both crops' cultures and livestock), averaged to 3.28% [27]. In addition, it accounts for 9.2% of the workforce, which confirms that it is one of the most important economic activities in relation to employment, particularly in rural areas. Geographically, silvoagricultural production in Chile, measured as participation of the sector in the regional GDP, is concentrated in the central-south part of the country, from Coquimbo to Los Lagos regions, 29–41° S [28].

In this scenario, an evaluation made by the Chilean Ministry of the Environment in 2019, as part of its commitments to CBD, states that main pressures or threats in connection to terrestrial ecosystems are both degradation and fragmentation. Land use change constitutes one of the main anthropic factors that has caused deterioration of the country's natural terrestrial ecosystems, together with the irregular cutting of forests and monoculture plantations based on exotic species. The agricultural industry, through forest clearance for placing grasslands and crops, is one of the most important causes of those changes. In

addition, both forestry and agricultural industries represent 96% of the total use of water at a national level [26].

Within this framework, we studied the allocation of IPPs to subsidize direct silvoagricultural activities (IPP-FFWs) in Chile, including amounts granted, the use supported, and their geographical application; we explored their eventual relationship with biodiversity loss patterns in the country in order to address international calls to both focus on subsidy accountability [29] as well as on indirect biodiversity loss drivers [16–18]. We also used it to address the national call to review instruments of promotion used by the state to face the challenges of an environmentally sustainable silvoagricultural production in Chile [28]. We hypothesize that, if Chile is adjusting to the requests to reduce detrimental subsidies, as committed in the CBD framework, IPP-FFWs ought to be diminishing or being reformulated through time.

## 2. Materials and Methods

Information about Instruments of Productive Promotion, offered by different governmental agencies in Chile, was gathered by reviewing scholarly publications, including a search in the Web of Science and Scielo platforms, using the term Instruments of Productive Promotion (IPPs), both in Spanish and English, and also by reviewing government official websites to obtain information and documents related to Instruments of Productive Promotion granted by them. The purpose was to generate a database with IPPs, including their ministerial dependence, nature of the instruments available, and activities funded. We focused on those instruments specifically oriented to financing field works (IPP-FFWs), which are those IPPs that have a direct territorial expression oriented to fund—among others—irrigation, forestation, and pasture improvement, all activities that might impinge upon land use and, hence, biodiversity. There are other governmental subsidies not used to fund field works, including—among others—technical assistance, capacity building, and associativity, that could indirectly have impacts on landscapes, which are not considered in this analysis since the main aim is to gather information on those instruments that directly impinge upon territories.

Subsequently, an official request of information associated with the granting of those IPP-FFWs was sent to the governmental agencies awarding them under the Chilean Transparency Law No. 20.285. Information requested through a questionnaire included records of beneficiaries since their first application or since the date information is recorded, amount of money granted yearly, and breakdown by administrative geographical unit. Finally, information received from governmental sources was analyzed from a territorial perspective in order to elucidate the geographical distribution of IPP-FFWs in Chile, including their representation by administrative region that was later compared to biodiversity features and trends in the country, as well as including identified drivers of biodiversity loss. Amounts received were corrected using the Consumers Price Index (IPC in Spanish) as a deflator and, then, were expressed in USD at the exchange rate of 1 USD = 738 Chilean Pesos.

## 3. Results

### 3.1. Instruments Offered by the Chilean State

There were 24 governmental agencies, offering a total of 131 Instruments of Productive Promotion (IPPs) oriented to different purposes, identified. Instruments devoted to training and capacity building are the most common (26%), followed by instruments used to finance field works and innovation (11% each). Other IPPs are allocated to foster associativity, commercial positioning events, credits with and without specifications, studies and research, insurance, and bonuses. Of the governmental agencies, four—Corporation for Promotion (Corfo), Institute for Agricultural Development (Indap), Prochile, and Service for Technical Cooperation (Sercotec)—encompass 63% of the IPPs identified (Table S1). From those 131 instruments identified, 15 (11%) are oriented to finance silvoagricultural activities or field work (IPP-FFWs) and are granted by six governmental agencies. Of those agencies, five are under the umbrella of the Chilean Ministry of Agriculture (Minagri).

Regarding the nature of the instruments, they mostly take the form of bonuses (10), either financing the total amount or by co-financing, and the other five (5) are offered as credits. In relation to beneficiaries, seven (7) IPP-FFWs are offered to rural family agriculture, four (4) are specifically oriented toward indigenous communities, two to farmers at large, two to owners of native forests, and one to owners of water rights. In terms of the purpose of the subsidy, irrigation and/or drainage, as well as water rights' purchase, have the higher number of IPP-FFWs available (n = 4), followed by forest management (n = 3), and all four other categories, including land purchase, pastures management, soil management, and improvement of silvoagricultural systems, have two each.

### 3.2. Amounts of IPP-FFWs Granted from 1976 to 2019

The request of information was answered by all six governmental agencies identified for granting IPP-FFWs (Table 1). The information received from the agencies came in different formats, with different levels of detail related to the criteria requested, spanning for different periods of time, and with different territorial representation. Moreover, despite public funds are being used, The Institute of Agricultural Development, Indap, did not provide information for four IPP-FFWs associated to credit, invoking Article 21 numeral 5 of Law 20.285 about Access to Public Information that refers to the protection of the private lives of users that have received these funds. In that context, information for only 11 out of 15 IPP-FFWs was received, including data for CNR Irrigation Promotion Law, Conadi 20a, 20b, and 20c, Conaf DL 701, Law.20.283, Indap PPSRF, PROM, and SIRSD-S, SAG Fund for the Improvement of Sanitary Heritage, as well as FIA Adaptation to Climate Change through Sustainable Development. Regarding the IPP-FFWs granting period, the information received ranged from 1976 to 2019. The instrument Law Decree 701, used for forestation activities, returned the longest records. In fact, from 1976 till 1994, there is only information for this instrument. Records of Conadi's IPP-FFWs appear in the year 1994. Indap provided data from 2005 onwards (Table 1). Considering that the reception of information was in August 2019 and that not all agencies provided data for that year, trend analyses were made with data until 2018.

From 1976 to 2019, Chilean governmental agencies granted ca. USD 3.25 billion associated with eleven IPP-FFWs, as per information provided. Promotion of irrigation was the most funded activity (ca. USD 906 million for Law of Irrigation promotion in 15 years), followed by forest management (ca. USD 753 million for Law decree 701 in 41 years), and land purchase for indigenous communities (ca. USD 733 million for Conadi 20b in 24 years). Altogether, these three IPP-FFWs account for USD 2.5 billion. The annual average of funding granted presented evidence that CNR Irrigation Promotion Law received ca. USD 57 million per year, followed by land purchase (Conadi 20b) with ca. USD 30 million, incentives to the agro-environmental sustainability of agricultural soils (SIRSD-S) with ca. USD 25 million, and LD 701 with ca. USD 18 million. All other IPP-FFWs received less than USD 10 million per year (Table 1).

IPP-FFWs allocation increased throughout the years (Figure 1a). Altogether, the total annual amounts allocated changed from ca. USD 40 million in year 2000 to more than USD 200 million in 2018, which quintupled the funding granted as IPP-FFWs in less than two decades. Public funds provided for irrigation and/or drainage and water rights' purchase, as well as land purchase, evidenced the most significant increase since 2000 (Figure 1b). Grouped by type of activity funded, IPP-FFWs directed to irrigation and/or drainage and water rights' purchase (Irrigation Promotion Law, PROM and Conadi 20c) amount to 32% of the total funding granted for which records were provided by governmental agencies; land purchase (Conadi 20a and 20b) amounted to 29%, and forest management (LD 701 and Law 20.283) amounted to 23%.

**Table 1.** Detail of IPP-FFWs granted by Chilean Agencies including amount in USD.

| Ministry | Agency | Name of the IPP-FFW | Initial Year | Final Year | USD | Annual Average |
|---|---|---|---|---|---|---|
| Agriculture | Institute for Agricultural Development (Indap) | System of Incentives to the Agro-environmental Sustainability of Agricultural Soils (SIRSD-S). | 2005 | 2019 | $366,801,158 | $24,453,411 |
| | | Program of Minor Irrigation Works (PROM). | 2009 | 2019 | $22,352,534 | $2,032,049 |
| | | Program of Supplementary Grasslands as Forage Resource (PPSRF). | 2011 | 2019 | $47,928,148 | $5,325,350 |
| | | Short Term Individual Link Credit to the Management of Supplementary Grasslands and Forage Resources. | Not Provided | Not Provided | Not Provided | Not Provided |
| | | Short Term Individual or Enterprises Link Credit to the Agro-environmental Sustainability of Agricultural Soils. | Not Provided | Not Provided | Not Provided | Not Provided |
| | | Long Term Individual or Enterprises Credit to the Management of Native Forest. | Not Provided | Not Provided | Not Provided | Not Provided |
| | | Long Term Individual or Enterprise Link Credit for Irrigation and Drainage. | Not Provided | Not Provided | Not Provided | Not Provided |
| | National Forestry Corporation (Conaf) | Law 20.283 on Recuperation of Native Forest and Forest Promotion. | 2010 | 2018 | $9,096,910 | $1,010,768 |
| | | Law Decree 701. Fix the legal regime of forestry land or preferably apt for forestation and establish norms of promotion. | 1976 | 2017 | $753,678,452 | $18,382,401 |
| | National Irrigation Commission (CNR) | Law of Irrigation Promotion. | 2004 | 2019 | $906,043,298 | $56,627,706 |
| | Foundation for Agricultural Innovation (FIA) | Adaptation to Climate Change through Sustainable Agriculture. | 2016 | 2018 | $11,726,263 | $3,908,754 |
| | Agricultural and Livestock Service (SAG) | Fund for the Improvement of Sanitary Heritage | 1999 | 2009 | $55,646,707 | $9,274,451 |
| Social Development and Family | National Corporation for Indigenous Development (Conadi) | Subsidy to the Acquisition of Land for Indigenous (20 a). | 1994 | 2019 | $227,943,875 | $10,361,085 |
| | | Land Purchase to Solve Land Problems (20 b). | 1994 | 2018 | $733,639,554 | $29,345,222 |
| | | Constitution, Regularization or Purchase of Water Rights or to Finance Works Oriented to Obtain that Resource (20 c). | 1994 | 2019 | $113,135,275 | $4,351,357 |
| | | Total | | | $3,247,983,175 | — |

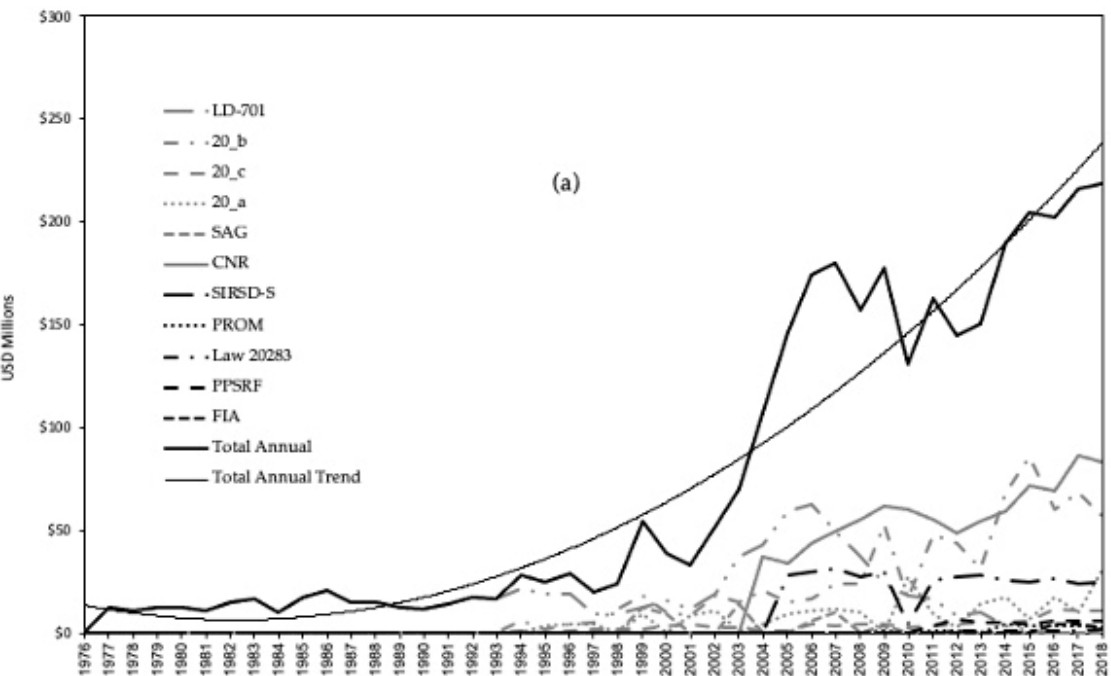

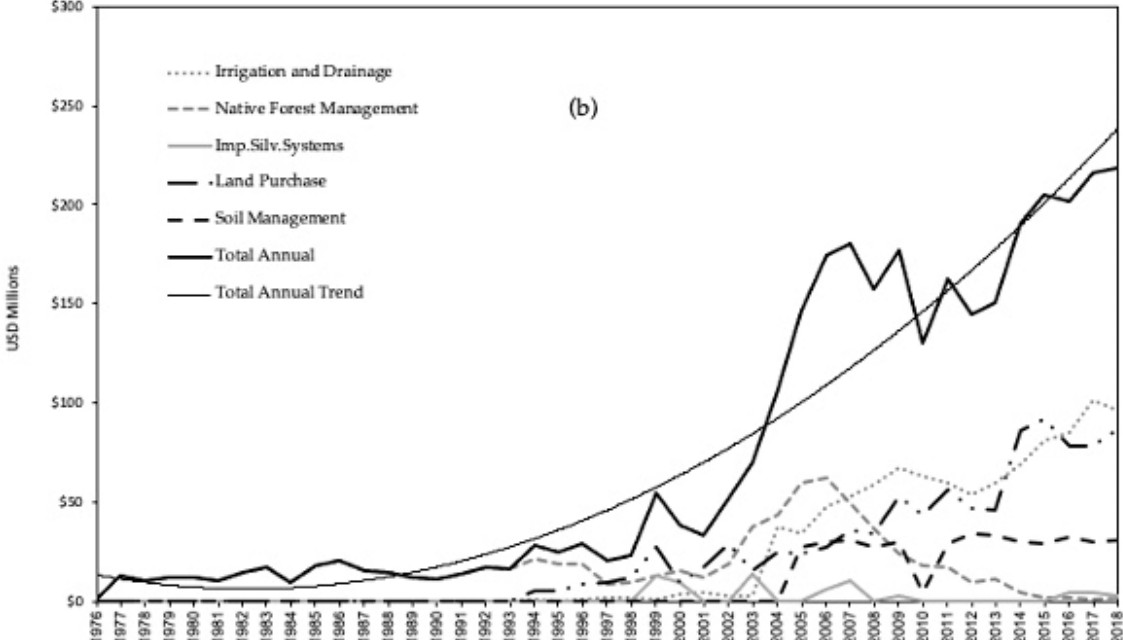

**Figure 1.** Funding granted for IPP-FFWs in Chile 1976–2018 in USD: (**a**) by instruments; (**b**) by type of activity. Instruments are: System of Incentives to the Agro-environmental Sustainability of Agricultural Soils (SIRSD-S); Program of Supplementary Grasslands as Forage Resource (PPSRF); Law of Irrigation Promotion (CNR); Program of Minor Irrigation Works (PROM); Constitution, Regularization or Purchase of Water Rights or to Finance Works Oriented to Obtain that Resource (20C_base); Law Decree 701; Law 20.283 on Recuperation of Native Forest and Forest Promotion; Subsidy to the Acquisition of Land for Indigenous (20A_base); Land Purchase to Solve Land Problems (20B_base), Adaptation to Climate Change through Sustainable Agriculture (FIA); Fund for the Improvement of Sanitary Heritage (SAG).

### 3.3. Geographical Distribution of IPP-FFWs

In geographical terms, distribution of the amounts granted for the 11 IPP-FFWs provides evidence that funds were allocated in all administrative regions of the country, with higher amounts concentrating from the Coquimbo Region (29° S) to the Los Lagos region (44° S) (Figure 2). The distribution of the higher amounts of IPP-FFWs allocated at the regional level (29–44° S) substantially overlap with Chilean biodiversity hotspots [30–32], concentrations of plant [33] and animal species richness, endemism and presence of threatened species [34], as well as concentrations of regional terrestrial ecosystem threat levels [35] (Figure 3).

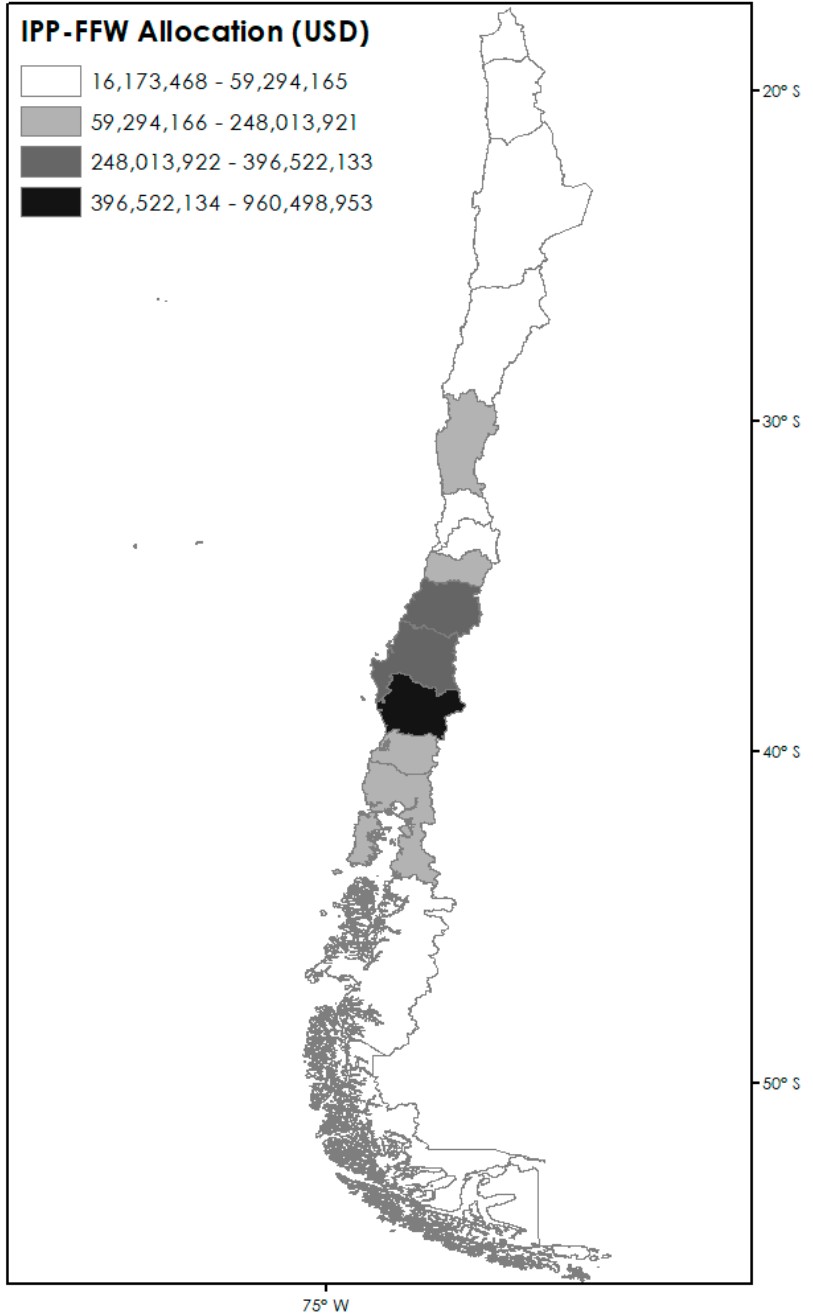

**Figure 2.** Regional distribution of IPP-FFWs funding granted in Chile from 1976 to 2019 in USD.

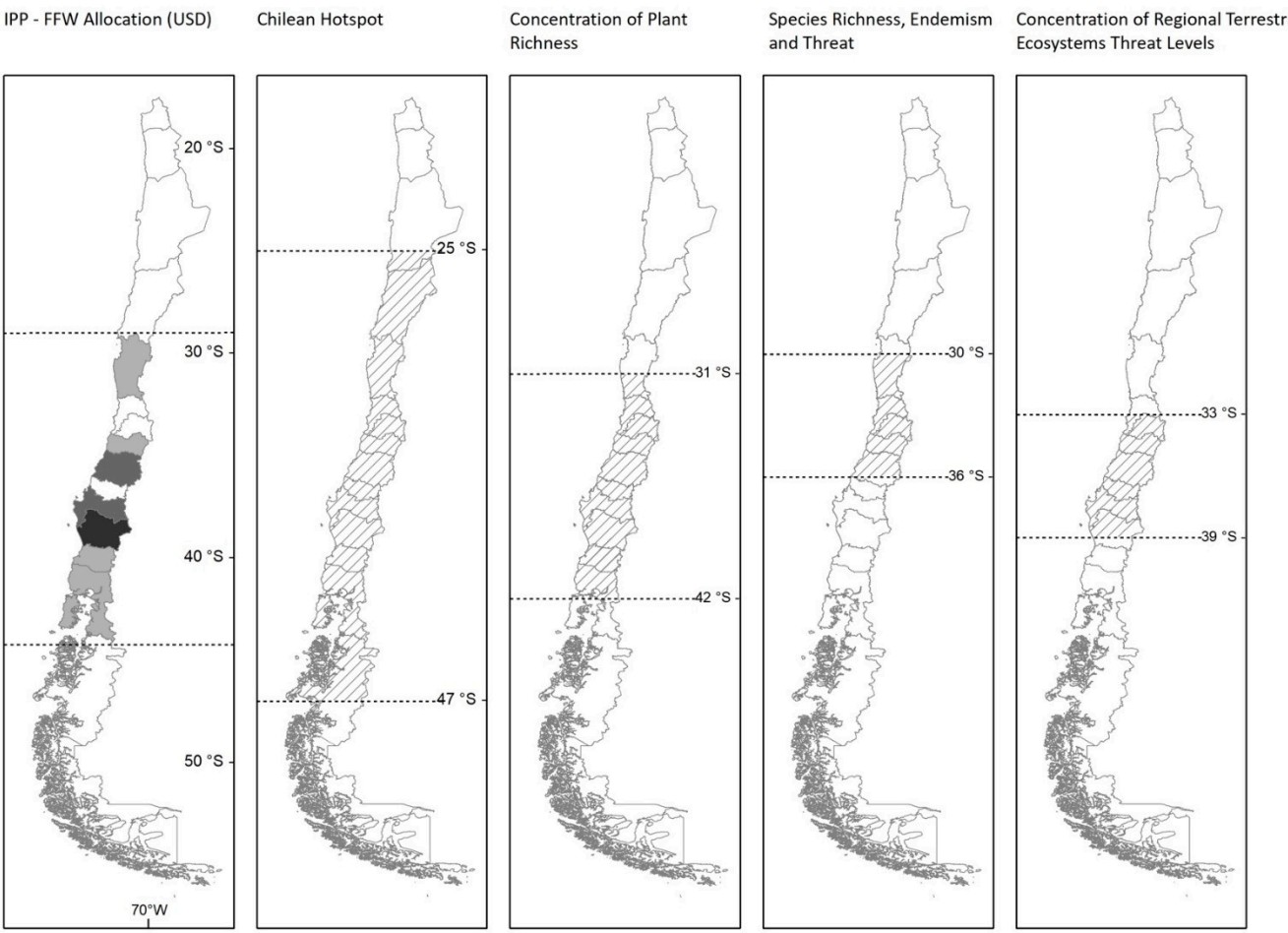

**Figure 3.** Regional distribution of IPP-FFWs funding granted in Chile, from 1976 to 2019 in USD, in relation to limits of relevant Chilean biodiversity characteristics. Different colors correspond to increasing funding granted per administrative region as shown in Figure 2.

## 4. Discussion

Through the Aichi targets, the international community agreed to stop and eliminate detrimental subsidies harmful to biodiversity, as well as to manage agricultural activities in a sustainable manner by 2020. However, an evaluation made by the Convention on Biological Diversity (CBD) in 2018 found evidence that only a few countries were both reducing and/or eliminating such subsides, and even less parties were identifying them systematically [36].

Chile, as signatory of the CBD, ought to be committed to fulfill those goals, but empirical evidence suggest otherwise. The analysis of information provided by governmental sources indicates that subsidies granted as IPP-FFWs have not diminished but, rather, increased. In fact, during the last two decades, the total annual amounts of IPP-FFWs have, at least, quintupled. This amount is underestimated since some information was not provided by the state agencies, due to access restrictions to data regarded as private, despite being public funds.

In that context, it must be considered that two out of the three most significant amounts of funds allocated by governmental agencies as IPP-FFWs have been used to promote irrigation and/or drainage, as well as forestation with exotic species. These activities have been associated with the main anthropogenic factors related to terrestrial ecosystems deterioration in Chile, as evidenced by governmental evaluations of the state of the environment [25,26].

Moreover, despite the fact that most of the IPP-FFWs are offered by agencies under the umbrella of the Ministry of Agriculture, there is neither an explicit policy of coordinated IPP-FFWs allocation in the silvoagricultural sector nor evidence of a coordinated sectoral efficiency assessment, which is in line with 2014 FAO analysis [23] that mentioned that, in Chile, there is not an explicit agro-environmental policy, and therefore, there is a need for such framework to be developed. This is especially important considering that the coordination and the linkage of instruments of promotion used by the state are considered as fundamental requisites to address the challenges of an environmentally sustainable silvoagricultural production [24]; subsidy programs are context-specific, so the way grants are allocated might lead to different outcomes [37]. This is relevant because the increase in tax revenue received by the state due to increased productivity could be used to cover biodiversity conservation needs [38]. In this context, a situation to be monitored is whether the recently enacted National Policy of Rural Development [39] will constitute the required framework [24].

Results of the geographical distribution of IPP-FFWs provide evidence that their allocation is concentrated in the central part of the country, an area that overlaps with one of the 25 biodiversity hotspots for conservation priorities recognized at the global level, due to its endemism and high degree of anthropogenic habitat impacts [30–32]. This region is also considered in six out of nine global biodiversity conservation priority schemes, highlighting its biological value [40], concentrating the highest richness of vascular flora at the family, genus, and species levels between 31–42° S [33], as well as over half of the endemic vertebrate species of the country between 30–36° S [34]. Further, according to the UICN classification of Chilean ecosystems, the area where subsidies allocation concentrates encompass most of ecosystems at risk in the country, including 8 categorized as Critically Endangered, 6 as Endangered, and 49 as Vulnerable [35]. In this context, it is also important to consider that contribution of the silvoagricultural sector to the regional Gross Domestic Product (GDP) is also concentrated in the same area (29–41° S) [28].

In addition, IPP-FFWs are not the subject of a strategic environmental assessment process, since this instrument has not yet been implemented at the practical level in the country [41]), nor are they the subject of environmental assessment [42,43]. So far, state granted activities that impinge upon land use changes and their associated drivers go unchecked in environmental regulatory terms, increasing the pressure upon the local biota.

The scenario depicted is possibly an underestimation of the subsidies allocated to activities that impinge upon biodiversity. First, only a fraction of the suite of subsidies in use in Chile do have information available. Second, there are other governmental subsidies not used to fund field works (i.e., capacity building), which were not included in our analysis but that might indirectly contribute to the execution of field activities supported by IPP-FFWs.

## 5. Conclusions

This study was aimed to generate basic information of silvoagricultural subsidies allocation in Chile as a way to address both national and international calls to focus on subsidy accountability to protect biodiversity. Findings provide evidence that IPP-FFWs allocation is increasing, that activities funded by those instruments are associated with the main anthropogenic factors related to terrestrial ecosystem deterioration in the country, and that allocations are concentrated in an area that overlaps with key biodiversity features, as well as the accumulation of threats of terrestrial ecosystems at the regional level, according to the IUCN criteria. Moreover, all these happen in the absence of a coordinated public policy of silvoagricultural subsidy allocation in the country.

Taking all of these into account, at the public policy level, we recommend working to generate basic information for coordinated decision-making on IPP-FFWs allocation, and to explore links between IPP-FFWs allocation and biodiversity status, to ascertain whether public funds could be acting as perverse incentives for its conservation. This is of the utmost importance considering that one of the pillars of the national goals on biological

diversity, for the period 2017–2030, is to incorporate elements to reduce the impacts on biodiversity and to incorporate criteria for its conservation in instruments of productive promotion (IPP-FFWs). Despite formal commitments, the challenge is still pending.

**Supplementary Materials:** The following supporting information can be downloaded at: https://www.mdpi.com/article/10.3390/su142215411/s1, Table S1. Agencies and number of IPPs granted by the Chilean State.

**Author Contributions:** C.P.: conceptualization; methodology; formal analysis, investigation; data curation; writing—original draft preparation; writing—review and editing. J.A.S.: conceptualization; methodology; formal analysis; writing—review and editing. All authors have read and agreed to the published version of the manuscript.

**Funding:** This research received no external funding.

**Institutional Review Board Statement:** Not applicable.

**Informed Consent Statement:** Not applicable.

**Data Availability Statement:** Not applicable.

**Acknowledgments:** We thank Mario Pezoa and Hernan Silva for their help in managing the databases. We appreciate comments by Cristian Estades, Miguel Castillo and Guillermo Donoso to an earlier version of this manuscript.

**Conflicts of Interest:** The authors declare no conflict of interest.

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
