# Peer review of "Subsidy Accountability and Biodiversity Loss Drivers: Following the Money in the Chilean Silvoagricultural Sector"

_sustainability, doi:10.3390/su142215411_

Round 1

Reviewer 1 Report

I have reviewed the manuscript dealing with the negative impacts of agricultural subsidies on environmental quality and biodiversity in Chile. While research aim is clear  and very important results have been generated, I  found a few areas of weaknesses that require revisions. A summary is provided below

1. The title is too long and needs to be reformulated so that it has relatively fewer words than the current one.  

2. The first reference sources ranging from number one (1) to number eight (8) are wrong and misleading. Please check your reference list at the end of the article and use authentic references and not ones given as instruction to authors regarding the formatting of references.  

3. In line 37-38, the authors are highlighting the negatives associated with agricultural subsidies on environmental quality and biodiversity which is fine. However, we need a balanced overview. To achieve this, please you must also indicate the positive value of agricultural subsidies in about 3 lines and thereafter proceed logically to summarise the negatives. 

4. I am not sure why is your manuscript lacking a conclusion and recommendation. If you don't conclude your research report, then you are loosing the opportunity to show out the significance and interpretation of your results. Similarly, you must indicate your recommendations on what needs to change or improve. 

5. The numbering of references 1-8 is highly flawed because these are not actually references that you have consulted. It seems they were included by mistake. 

Sincerely,

Reviewer of Manuscript 

Reviewer 2 Report

Although the article is valuable, I have a few comments. I think making the following corrections will raise the scientific value of the article.

Comments:

- Despite the interesting problem, subsection 3 is a kind of "statistical enumeration". This section is 8 pages long, 6 of which are tables and figures. Section 3.3 it's just 6 lines of text.

The entire section 3 lacks valuable statistical analyses (let alone econometric analyses). The empirical section should be rated low. Already adding a trend line in figure 1 (e.g. in excel) and commenting on the result, would have increased the empirical value
- throughout the paper it is difficult to check the quality of the citations because there is bad numbering of the cited publications. The first 8 items in "References" should be discarded.
- in my opinion there are few recent literature items. Publications from before 2015 clearly dominate.
- due to the length of the "4. Discussion" section, it is worth separating the "Conclusion" section
- Please add a few limitations sentences (in the Conclusions/Discussion section). What, according to the authors, may be a limitation of such research.

Minor:
- Key words should rather be written in lower case,
- Tables 1 and 2 should be adapted to the publishing requirements (e.g. font type),
- Page numbers are mixed up,

Good luck,

Round 2

Reviewer 1 Report

I am happy that most of my comments have been duly addressed by the authors 

Reviewer 2 Report

I accept the explanations and appraciate the changes. I think the paper is much better in its present form.